# Tensile and Compressive Properties of Woven Fabric Carbon Fiber-Reinforced Polymer Laminates Containing Three-Dimensional Microvascular Channels

**DOI:** 10.3390/polym16050665

**Published:** 2024-02-29

**Authors:** Ziqian An, Xiaoquan Cheng, Dafang Zhao, Yihao Ma, Xin Guo, Yujia Cheng

**Affiliations:** 1School of Aeronautic Science and Engineering, Beihang University, Beijing 100191, China; anziqian@buaa.edu.cn (Z.A.); gxdfxxx@163.com (X.G.); yujia.cheng@buaa.edu.cn (Y.C.); 2Aviation Industry Corporation of China, Ltd. (AVIC) Manufacturing Technology Institute, Beijing 100024, China; zhaodf003@avic.com; 3Research Institute of Navigation and Control Technology, China North Industries Group, Beijing 100089, China; mayihao@buaa.edu.cn

**Keywords:** self-healing composites, woven fabric CFRP, microvascular, experimental test, finite element analysis

## Abstract

Microvascular self-healing composite materials have significant potential for application and their mechanical properties need in-depth investigation. In this paper, the tensile and compressive properties of woven fabric carbon fiber-reinforced polymer (CFRP) laminates containing three-dimensional microvascular channels were investigated experimentally. Several detailed finite element (FE) models were established to simulate the mechanical behavior of the laminate and the effectiveness of different models was examined. The damage propagation process of the microvascular laminates and the influence of microvascular parameters were studied by the validated models. The results show that microvascular channels arranged along the thickness direction (z-direction) of the laminates are critical locations under the loads. The channels have minimal effect on the stiffness of the laminates but cause a certain reduction in strength, which varies approximately linearly with the z-direction channel diameter within its common design range of 0.1~1 mm. It is necessary to consider the resin-rich region formed around microvascular channels in the warp and weft fiber yarns of the woven fabric composite when establishing the FE model. The layers in the model should be assigned with equivalent unidirectional ply material in order to calculate the mechanical properties of laminates correctly.

## 1. Introduction

Due to the poor interlaminar performance of fiber-reinforced polymer composite laminates and the shortcomings of existing non-destructive testing and repair methods for composite structures [1], self-healing structural polymers and fiber-reinforced composites have been proposed, which can be divided into intrinsic and external approaches [2]. The intrinsic self-healing system is mainly based on the reversible chemical reaction of the matrix material itself, which does not affect the structural integrity but is only suitable for repairing small damages or scratches [3]. The external approaches, inspired by the self-healing characteristics of organisms after injury, use microcapsules, hollow fibers, or micro channels to transport healing agents to promptly repair damages [4]. Among these approaches, microvascular self-healing is more promising for it allows multiple efficient repairs of delamination damage or matrix cracks [5,6,7,8,9]. It can effectively reduce maintenance costs, improve safety, and extend structure service life if implemented.

Luterbacher et al. [10] incorporated microvascular self-healing into a composite skin-stringer structure to deliver the healing agent and found that the structural performances could be fully restored by using microvascular self-healing to repair the debonding interface between the stringer and skin panel. Sakurayama et al. [11] conducted impact and compression tests on composite stiffened panels containing microvascular networks and repaired the impact damage using them. The results showed that the repaired stiffened panel could recover 50% of its compression strength compared to the unrepaired specimens. These all confirm the potential of microvascular systems under practical conditions. Additionally, there are relatively mature technical routes in terms of manufacture processes and healing agents, such as the vaporization of sacrificial component (VaSC) method [7] and the epoxy resin system [12]. However, the microvascular channels can also be regarded as initial damage that affects the mechanical properties of the structure. Therefore, it is necessary to thoroughly study the mechanical performance of laminates containing microvascular channels so as to determine appropriate design parameters in actual structures.

A number of experimental studies have now been conducted to address the issue of the mechanical performance of microvascular composites. Kousourakis et al. [13] tested the tensile and compressive properties of laminates containing micro channels located in the mid-plane of the laminate. As the diameter of the microvascular channels increased from 0.3 mm to 3 mm, the strength and stiffness of the specimens with longitudinally oriented channels decreased by less than 10%, while the performance of the specimens with transversely oriented channels decreased significantly, with a maximum reduction in tensile strength of 50%. The main reason for this significant performance loss was the bending of fibers around the channels which resulted in a change in the stress state. Devi et al. [14] also reached similar conclusions. Saeed et al. [15] conducted three-point bending and short beam strength tests on laminates containing in-plane microvascular channels and found that both the bending strength and short beam strength of the specimens linearly decreased as the diameter of the channels increased. With a channel diameter of 1.5 mm, the short beam strength decreased by about 33%, and the bending strength decreased by about 15%. Coppola et al. [16] investigated the tensile properties and damage propagation of 3D orthogonally woven glass fiber composites containing straight and undulating wave-shaped micro channels and found that reductions in strength and modulus only occurred when channels distorted the fiber architecture. Norris et al. [17] found that cutting the fibers around the channel can prevent the formation of a resin-rich region, but this will lead to a more significant decrease in the mechanical performance of the laminate.

Some researchers have also attempted to conduct studies using finite element methods. Nguyen and Orifici [18] first conducted experiments on laminates containing micro channels with a diameter of 0.68 mm. They found that the tensile stiffness of the laminate perpendicular to the microvascular channels could decrease by up to 7.5%, and the compressive strength could decrease by 4.9%, while the performance decrease along the direction of the channel was not significant. They further established a representative volume element (RVE) model of the microvascular channel. The composite plies were modeled using continuous shell elements, and the two-dimensional Hashin criterion was used to determine the damage of the composite material. The resin-rich region was considered and the numerical results of mechanical performance and failure modes were in good agreement with the experimental results. Huang et al. [19] established a plane strain model, while Shawk et al. [20], Demiral et al. [21], and Zhao et al. [22] established three-dimensional models to study the influence of in-plane microvascular channels on different mechanical properties of laminates. Ran et al. [23] also considered the variation in fiber volume fraction in the area around the microvascular channels where fibers are bent in an FE model. Compared to models that do not consider this factor, the calculated results of laminate strength and stiffness were more accurate.

It is evident that current researchers primarily focus on laminates with a one-dimensional microvascular channel arranged between composite layers. A few studies have demonstrated research on three-dimensional microvascular channels but lack simulation analysis. Delamination damage can occur at any position within the laminate during the structures’ manufacturing and service period [24]. In-plane micro channels can only repair damage between specific layers, and arranging channels in multiple layers would lead to a significant decline in the laminate’s mechanical performance. Therefore, in a three-dimensional microvascular configuration, the in-plane microvascular channels are used to transport healing agents and the z-direction microvascular channels are used to repair delamination damage at different positions. This may be the way to make self-healing structures available for engineering applications. In order to provide a reference and basis for the design of microvascular composite structures, studies on the mechanical performance of laminates with such microvascular configurations should be conducted.

In this paper, the tensile and compressive properties of woven fabric CFRP laminates containing three-dimensional microvascular channels were investigated experimentally. New detailed finite element models with a resin-rich region and variations in fiber volume fraction around the z-direction microvascular based on the actual structure were established, which were employed to study the damage propagation of laminates under tensile and compressive loads. The effects of microvascular parameters, including diameter, spacing and volume fraction, on the tensile and compressive properties of the laminates were discussed using the FE model. Finally, the design criteria for microvascular self-healing composite structures were summarized based on the parameter study results.

## 2. Experiment

### 2.1. Material

The laminates in this study were all made of CF3031 carbon fiber fabric and 5284 epoxy resin, with the mechanical properties listed in Table 1. The materials and the nominal properties were all provided by AVIC Manufacturing Technology Institute.

### 2.2. Specimen Design and Manufacture

The tensile and compression specimens containing microvascular channels were designed according to the ASTM D3039 [25] and D6641 [26] standards. The specimens contained two parallel, three-dimensional microvascular channels, as illustrated in Figure 1. The layup of the specimens was [(0,90)/±45/(0,90)/±45/(0,90)] s, with a nominal thickness of 0.25 mm per layer. The width of the specimens was 24 mm, and the spacing of the z-direction channels was 12 mm. The diameter of the channel was 0.5 mm, with the in-plane channels located two layers beneath the surface of the specimen, as illustrated in Figure 2. For compression specimens, care was taken during preparation to ensure the presence of z-direction channel within the gage section. Blank specimens were also prepared as controls.

In the manufacturing procedure, the carbon fiber-woven fabric was manually laid to form a preform and polylactic acid (PLA) threads were sewn into the preform at pre-determined intervals. After this, the vacuum-assisted resin infusion (VARI) process was used for resin impregnation and curing. Finally, the VaSC method was used when the PLA threads were evaporated in an oven, leaving hollow channels. In order to make their decomposition temperature much lower than the glass transition temperature of the composite matrix resin, some catalyst was added into the PLA threads.

### 2.3. Mechanical Testing

Tensile and compression tests were conducted on an INSTRON-8801 testing machine (Norwood, MA, USA) in a standard laboratory environment (23 ± 2 °C, 50 ± 10% relative humidity), using displacement control loading at a rate of 2 mm/min. The width and thickness of the gage section of each specimen were measured three times before testing, and the average values were taken. Five microvascular specimens and five blank specimens were tested in both experiments. The microvascular specimens for tensile and compression test were numbered XT-P-1~5 and XC-P-1~5, while the blank specimens were numbered XT-C-1~5 and XC-C-1~5 accordingly.

The tensile test was conducted according to ASTM D3039; strain in the longitudinal and transverse directions of the specimen was measured using two extensometers. The stiffness of the specimen was calculated using data from the longitudinal extensometer within the range of 1000 με to 3000 με. Extensometers were removed when the longitudinal strain was 5000 με, then the specimen was stretched to failure. The failure load and mode were recorded. The setup of the test is shown in Figure 3.

The compression test was conducted according to ASTM D6641. Four strain gauges were used on the gage section to measure the strain of the specimen. The stiffness of the specimen was calculated using the gauges’ data within the range of 1000 με to 3000 με. The specimen was loaded until failure, and the failure load and mode were recorded. The location of the strain gauges and the setup of the test are shown in Figure 3.

All specimens were determined to have failed when visible fracture occurred and the load of the testing machine rapidly decreased by more than 30%.

### 2.4. Result and Analysis

Test results are shown in Table 2. The tensile stiffness and strength of the specimens with microvascular channels reduced by 6.1% and 11.9% compared with the control group, while the compressive strength reduced by 10.3% but the stiffness increased by 2.1%. The stress–displacement curves of the tensile specimens and stress–strain curves of the compression specimens are shown in Figure 4. It can be observed that the microvascular has an evident effect on the strength of the laminates, but a relatively smaller effect on the stiffness. When the strain is small, the curves of the two types of specimens are very close to each other. The increased stiffness of the compression specimens with microvascular channels may be due to the dispersion of the material properties, which has a greater effect than that of the microvascular channels.

The failure modes of the tensile specimens are shown in Figure 5a. Three of the specimens failed at the middle of the gage section. Figure 5b shows the fracture section captured by optical microscopy and the z-direction channels are visible, suggesting that the z-direction microvascular channels are found at key sections of these specimens when carrying tensile load. Specimens XT-P-1 and XT-P-2 failed near the grip section where stress concentration existed, but their strengths were not significantly lower than others.

For compression specimens, the primary failure mode was brooming fracture in the middle of the gage section where the z-direction microvascular channel was located, as shown in Figure 5c; however, it was difficult to determine the z-direction channel due to the extensive damage in the matrix of the compression specimens.

When bearing loads, the z-direction microvascular channel causes stress concentration around it, thereby reducing the strength. However, since the microvascular channel only causes a slight bending of fibers in local areas, which is also a feature of the woven fabric material itself, the fibers are not cut and the total fiber volume does not change, so that the stiffness of laminates is barely affected.

## 3. Finite Element Model

To further investigate the failure mechanism of laminates containing microvascular channels and to conduct parametric studies, it is necessary to establish a detailed finite element model; however, modeling the whole structure is too hard to achieve. It is advisable to consider in-plane and z-direction microvascular channels separately. The mechanical properties of laminates containing in-plane microvascular channels have been the focus of some studies, in which it was observed that the mechanical properties of the laminates along the direction of the in-plane channel were less affected [13,14,16,18]. Also, the experimental results in Section 2 indicate that the z-direction of microvascular channels is a critical position. Therefore, the model will focus on the z-direction of microvascular channels and its influence on the mechanical properties of the laminate. Finite element analysis was performed on ABAQUS 6.14 software.

### 3.1. Model Generation

#### 3.1.1. Resin-Rich Region

In laminates, the resin-rich region around microvascular channels can cause stress concentration or fiber bending, which are key factors affecting the mechanical performance of the laminate. It is important to consider the resin-rich region in the FE model. Ma et al. [27] developed a numerical method to predict the shape of the resin-rich region around microvascular channels, and this paper refers to this method to determine the length of the resin-rich region.

The z-direction microvascular channels in woven fabric composite will lead to the formation of two perpendicular resin-rich regions within the warp and weft fiber yarns in one ply. Since the forming process involves stitching sacrificial lines into the preform before resin impregnation, the fibers are still continuous. Typical intralaminar resin-rich region in the current specimens are shown in Figure 6.

Some specimens were cut to measure the length of the resin-rich region. The measurement results were compared with the numerical results as illustrated in Figure 7. The measurement results show obvious dispersion, which is primarily due to the bending of fiber in the fabric material, and so the complete resin-rich region might not be observable on certain thickness sections. By adjusting the parameter values in the numerical methods, the calculated results exceeded 95% of the measured values. The lengths of the resin-rich region used in the FE models are presented in Table 3.

#### 3.1.2. Variation of Fiber Volume Fraction

The fiber volume fraction around the microvascular channel will change as the fibers are pushed away, and the mechanical properties of the local material will be affected. The authors of [23] illustrated the necessity of considering this factor in in-plane microvascular models. This paper also considers this factor in the model of the z-direction microvascular channels first, and then compares different modeling approaches without it.

Since there are two resin-rich regions in the warp and weft yarns in one layer, it is necessary to model them separately. This paper assumed that the fabric layer was divided into warp and weft sublayers in the model, and each sublayer was treated as an equivalent unidirectional ply. By assigning appropriate material properties, the mechanical performances of the combination of two sublayers remained the same as those of the original fabric material, and the properties of the sublayers and fabric layers satisfy the following relationships:(1)$$E_{L}=E_{T}=\frac{1}{2}\left(E_{11}+E_{22}\right)$$
(2)$$\nu_{L}=\nu_{T}=\frac{\nu_{21}E_{11}+\nu_{12}E_{22}}{E_{11}+E_{22}}$$
(3)$$G_{\mathrm{LT}}=G_{12}$$
(4)$$X_{\mathrm{Lt}}=X_{\mathrm{Tt}}=\frac{1}{2}\left(X_{t}+\frac{X_{t}}{E_{11}}E_{22}\right)$$
(5)$$X_{\mathrm{Lc}}=X_{\mathrm{Tc}}=\frac{1}{2}\left(X_{c}+\frac{X_{c}}{E_{11}}E_{22}\right)$$
where the subscripts L and T represent the longitudinal and transverse direction of the fabric. *E*_L_, *E*_T_, *ν*_L_, *ν*_T_, *G*_LT_, *X*_Lt_, *X*_Tt_, and *X*_Lc_, *X*_Tc_ are the elastic modulus, Poisson’s ratio, shear modulus, tensile strength, and compression strength of the woven fabric. *E*_11_, *E*_22_, *ν*_12_, *ν*_21_, *G*_12_, *X*_t_, and *X*_c_ are the elastic modulus, Poisson’s ratio, shear modulus, tensile strength and compression strength of the unidirectional ply. The out-of-plane properties of the two materials are considered the same.

In order to ensure that the simulation results of the fabric material are consistent with the nominal values, it is necessary to apply correction factors when calculating the modulus and strength parameters of the unidirectional ply based on the properties of the fibers and resin. Referring to the Chamis [28] model, the calculation formulas are as follows:(6)$$E_{11}=\alpha_{1}\left(V^{f}E_{11}^{f}+V^{m}E^{m}\right)$$
(7)$$E_{22}=\alpha_{2}\frac{E_{22}^{f}E^{m}}{E_{22}^{f}-\sqrt{V^{f}}\left(E_{22}^{f}-E^{m}\right)}$$
(8)$$\nu_{12}=\alpha_{3}\left(V^{f}\nu_{11}^{f}+V^{m}\nu^{m}\right)$$
(9)$$G_{12}=\alpha_{4}\frac{G_{12}^{f}G^{m}}{G_{12}^{f}-\sqrt{V^{f}}\left(G_{12}^{f}-G^{m}\right)}$$
(10)$$G_{23}=\alpha_{5}\frac{G_{23}^{f}G^{m}}{G_{23}^{f}-\sqrt{V^{f}}\left(G_{23}^{f}-G^{m}\right)}$$
(11)$$X_{t}=\beta_{t}\left(V^{f}\sigma_{f}+V^{m}\frac{\sigma_{f}}{E_{11}^{f}}E^{m}\right)$$
(12)$$X_{c}=\beta_{c}\left(V^{f}\sigma_{f}+V^{m}\frac{\sigma_{f}}{E_{11}^{f}}E^{m}\right)$$
where the subscripts f and m represent the fiber and resin. *V*^f^, *V*^m^ are the fiber and resin’s volume fraction. *E*^m^, *G*^m^, and *ν*^m^ are the elastic modulus, Poisson’s ratio, and shear modulus of the resin. $E_{11}^{f}$, $E_{22}^{f}$ are the elastic moduli in the longitudinal and transverse directions of the fiber. $\nu_{12}^{f}$, $G_{12}^{f}$, and $G_{23}^{f}$ are the Poisson’s ratio and shear modulus of the fiber. *σ*_f_ is the fiber’s strength. *α* and *β* are correction factors.

The properties of the fibers and resin used in the research are shown in Table 4, and the fiber volume fraction is 55%. The correction factors must be adjusted to ensure that the failure strain in the fiber direction of the unidirectional ply matches that of the actual fabric material, as indicated in Table 5. The calculated properties of the equivalent unidirectional ply are presented in Table 6. It is assumed that the transverse tensile and compressive strengths of the unidirectional ply are same as the strengths of the matrix. The material properties of the unidirectional ply used in the following models are all calculated using the method described in this section.

### 3.2. Failure Criteria and Material Property Degradation

#### 3.2.1. CFRP Material

Three-dimensional Hashin failure criteria [29], Chang fiber-shear failure criteria [30], and Ye delamination failure criteria [31] were employed to predict the different damage modes in the laminate, which are explained in detail as follows:

Fiber failure:(13)$${\left(\frac{\sigma_{11}}{X_{T}}\right)}^{2}+{\left(\frac{\tau_{12}}{S_{12}}\right)}^{2}+{\left(\frac{\tau_{13}}{S_{13}}\right)}^{2}\geq 1\;(\sigma_{11}\geq 0)$$
(14)$${\left(\frac{\sigma_{11}}{X_{C}}\right)}^{2}\geq 1\;(\sigma_{11}<0)$$

Matrix failure:(15)$${\left(\frac{\sigma_{22}}{Y_{T}}\right)}^{2}+{\left(\frac{\tau_{12}}{S_{12}}\right)}^{2}+{\left(\frac{\tau_{23}}{S_{23}}\right)}^{2}\geq 1\;(\sigma_{22}\geq 0)$$
(16)$${\left(\frac{\sigma_{22}}{Y_{C}}\right)}^{2}+{\left(\frac{\tau_{12}}{S_{12}}\right)}^{2}+{\left(\frac{\tau_{23}}{S_{23}}\right)}^{2}\geq 1\;(\sigma_{22}<0)$$

Fiber–matrix shear failure:(17)$${\left(\frac{\sigma_{11}}{X_{C}}\right)}^{2}+{\left(\frac{\tau_{12}}{S_{12}}\right)}^{2}+{\left(\frac{\tau_{13}}{S_{13}}\right)}^{2}\geq 1\;(\sigma_{11}<0)$$

Delamination:(18)$${\left(\frac{\sigma_{33}}{Z_{T}}\right)}^{2}+{\left(\frac{\tau_{13}}{S_{13}}\right)}^{2}+{\left(\frac{\tau_{23}}{S_{23}}\right)}^{2}\geq 1\;(\sigma_{33}\geq 0)$$
(19)$${\left(\frac{\sigma_{33}}{Z_{C}}\right)}^{2}+{\left(\frac{\tau_{13}}{S_{13}}\right)}^{2}+{\left(\frac{\tau_{23}}{S_{23}}\right)}^{2}\geq 1\;(\sigma_{33}<0)$$where *σ*_11_, *σ*_22_, and *σ*_33_ are normal stress components along the longitudinal, transverse, and thickness directions, respectively. *τ*_12_, *τ*_13_, and *τ*_23_ are shear stress components. *X*_T_ and *X*_C_ are tensile and compressive strengths along the longitudinal direction. *Y*_T_ and *Y*_C_ are tensile and compressive strengths along the transverse direction. *Z*_T_ and *Z*_C_ are tensile and compressive strengths along the thickness direction. *S*_12_, *S*_13_, and *S*_23_ are shear strengths. The properties of unidirectional CFRP were obtained by the method presented in Section 3.1.2 and listed in Table 1 and Table 6.

Once the stress state of an element satisfies any of the above failure criteria, stiffness parameters of the element will be degraded to a certain value according to the degradation rules proposed by Camanho and Matthews [32] as listed in Table 7.

#### 3.2.2. Resin

For the resin-rich region, the parabolic criterion [33] was used to determine the initiation of resin damage:(20)$$\frac{3J+I\left(S_{\mathrm{mc}}-S_{\mathrm{mt}}\right)}{S_{\mathrm{mc}}S_{\mathrm{mt}}}=1\;(I\geq 1)$$
(21)$$-\frac{3J=I\left(S_{\mathrm{mc}}-S_{\mathrm{mt}}\right)}{S_{\mathrm{mc}}S_{\mathrm{mt}}}=1\;(I<1)$$
where *S*_mt_ and *S*_mc_ are the unidirectional tensile and compressive strength of resin as listed in Table 4. *I* and *J* are defined as follows:(22)$$I={\tilde{\sigma }}_{m1}+{\tilde{\sigma }}_{m2}+{\tilde{\sigma }}_{m3}$$
(23)$$J=\frac{1}{6}\left[{\left({\tilde{\sigma }}_{m1}-{\tilde{\sigma }}_{m2}\right)}^{2}+{\left({\tilde{\sigma }}_{m1}-{\tilde{\sigma }}_{m3}\right)}^{2}+{\left({\tilde{\sigma }}_{m2}-{\tilde{\sigma }}_{m3}\right)}^{2}\right]$$
where ${\tilde{\sigma }}_{m1}$, ${\tilde{\sigma }}_{m2}$, and ${\tilde{\sigma }}_{m3}$ are principle stress components. The stiffness parameters of failure elements are degraded according to the relation of $E^{m}=0.2E^{m},\mu^{m}=0.2\mu^{m}$. The failure criteria and material property degradation rules of CFRP and resin are defined in the VUMAT subroutine of ABAQUS.

### 3.3. Model Details

The model with z-direction microvascular channels that considers the resin-rich region, fiber bending, and variations in material properties is shown in Figure 8, referred to as model A. In this model, it is assumed that the fiber volume fraction linearly decreases from the edge of the channel to the surrounding area, while ensuring the conservation of the total fiber volume. Different colors in the elements represent different material properties. This was accomplished by using Python script. The coordinates of each node were read, and the fiber volume fraction was calculated based on the distance from the center of the element to the center of the channel. Then, the material parameters were calculated according to Formulas (6)~(12) and assigned to the corresponding elements.

Symmetric models were established to save computational time. The tensile model only includes the gage section, while the compression model does not include reinforcement tabs. The z-direction microvascular structure was tied to the overall structure. One end of the model was fixed, the other was coupled with a reference point where the displacement load is applied, and the reaction force was extracted. In the tensile model, the displacements of the side nodes were extracted to calculate the stiffness, while in the compression model, the strain at the center of the surface was extracted for the same purpose, consistent with the experimental measurement method.

The fiber bending region and resin-rich region near the channel were meshed finely to investigate the damage propagation in more detail with a maximum element size of 130 μm in the x- and y-directions. The main element type was C3D8R, with a few C3D6 elements in the resin-rich region, and the total number of elements in tensile and compression models were 110,344 and 104,616, respectively. The models of the laminates, including boundary conditions, loading condition, and nodes for result output, are shown in Figure 9.

To investigate the necessity of this modeling approach, several other z-direction microvascular models were also established for comparison. Model B did not consider the variation in fiber volume fraction compared to model A. Model C did not divide the fabric layer into two unidirectional ply and applied fabric material directly compared to model B. These two models had the same mesh as model A. Model D did not consider the resin-rich regions, bending of fibers near the channel, and variation in fiber volume fraction. The model contained 45,292 elements and also applied fabric material properties. This modeling process is simple but equivalent to cutting the fibers and reducing the total fiber volume. The control models are shown in Figure 10.

### 3.4. Validation and Comparison of the FE Models

The tensile and compressive performances of the microvascular laminates calculated by each model, along with their comparison with experimental results, are listed in Table 8. It shows that since model A considers the variation in the fiber volume fraction around the microvascular channel, which results in higher material strength near the channel, the strengths of the laminate are slightly greater than those of model B. However, the results of model B are still very close to the experimental data. The strength results of model C and D are significantly lower than the experimental results, indicating that these modeling approaches do not accurately reflect the actual performances of the structure. In FE models, the element layers with a longitudinal resin-rich region bear greater load because of higher stiffness and the total load reaches peak value when they fail. So, the strengths of the whole model mainly depend on the properties of these layers, which explains why model C has a larger error with a fabric material assigned. In model D, the fibers are cut and stress concentration is severer, which is inconsistent with reality. The details are discussed in the following section. As for the stiffness performances of the laminates, since the impact of the microvascular channels is minimal, the simulation results of all models are relatively close.

The damage configurations of test and simulation results are listed in Table 9. The tensile specimen was polished to observe the channels and the z-direction channel is circled in red. As can be seen, the numerical results all exhibit lateral tensile or compressive failures. The failure fractures of models A and B are relatively more jagged and closer to the test results, further demonstrating the better rationality of models A and B.

Based on the above results, it can be concluded that when modeling the z-direction microvascular channel in woven fabric material, the resin-rich regions formed within the warp and weft fibers and the bending of fibers should be considered. Furthermore, the fabric layer should be equivalently modeled as two unidirectional plies. It is not necessary to consider the variation in the fiber volume fraction around the microvascular channel, which simplifies the modeling process appropriately.

### 3.5. Damage Mechanism

Figure 11 presents the load–displacement curves of the microvascular laminates numerically in model A, as well as the damage propagation process in different layers of the laminate.

During the tensile process, fiber damage first appeared around the microvascular channel in the internal 0° layers of the laminate at point A (19.5 kN). At point B (20.4 kN), resin damage appeared around the microvascular channel in the internal 90° layers, and extended to the entire resin-rich region at point C (21.4 kN). Before reaching the peak load, the fiber damage in the internal 0° layers continued to extend towards the edges of the laminate, while the surface layers showed no significant damage. After reaching the peak load at point E (23.2 kN), extensive fiber damage rapidly occurred in all layers until the specimen ultimately failed.

During the compression process, fiber damage first appeared around the microvascular channel in the internal 0° layer at point A (20.1 kN), and damage in the resin-rich region around in the 90° layers also occurred. The damage had almost completely penetrated the resin-rich region by point B (21.9 kN). Before reaching the peak load, the fiber damage in the internal layers continued to extend from the edge of the channel towards the laminate edges, with no significant damage in the surface layers. After reaching the peak load at point D (25.4 kN), extensive fiber damage occurred near the channel and at the tips of the resin-rich region in the internal 0° layers, while significant damage also appeared in the surface layers, leading to the final failure of the specimen.

Figure 12 presents the *S*_11_ stress maps around the microvascular channel in the 0° layer during the loading process, with the laminate under tensile or compression load of 70 MPa and showing no damage. For comparison, the stress maps calculated by model D without resin-rich regions are also presented under the same load conditions. When the resin-rich regions are present, the stress concentration factor around the microvascular channel is approximately 1.44, whereas it is about 2.3 when not considering the resin-rich region. It can be seen that although the presence of resin-rich regions leads to stress concentration, their impact is significantly less than directly drilling holes in the laminate, resulting in a relatively smaller loss in the strength of the laminate.

## 4. Parameter Study

From Section 3, it is known that model B can calculate the mechanical properties of the laminates accurately and the modeling approach is easier than with model A. Therefore, this modeling method was utilized to investigate the influence of microvascular parameters on the mechanical properties of the laminates.

### 4.1. Diameter

The materials and layup sequence are the same as those in Section 3, with channel diameters ranging from 0.1 mm to 1 mm. This range is commonly used in current research, for smaller diameters are not conducive to the flow of the repair agent while larger diameters affect the laminate’s mechanical performances too much, which is impractical. The spacing of channels, *S*, was set to 8, 12, and 18 mm. The changes in the stiffness and tensile/compressive strengths of the laminates with varying channel diameters are shown in Figure 13. All calculation results presented as a percentage relative to the results of the control model.

It can be observed that the mechanical properties of the microvascular laminates are generally lower than those of the control laminates. However, within the diameter range of 0.1~1 mm, the decrease in laminate stiffness is less than 3%, with a slightly increasing rate of decrease, while the strength of the laminates decreases approximately linearly with channel diameters. Microvascular channels with a spacing of 8 mm and a diameter of 1 mm can reduce the tensile and compressive strength of the laminates by about 15%.

### 4.2. Spacing

With microvascular channel diameters *D* set to 0.1, 0.5, and 1 mm, the changes in the stiffness and tensile/compressive strengths of the laminates with varying spacing of channel are shown in Figure 14. It can also be seen that within the current range of design parameters, the stiffness of the laminates is minimally affected by the microvascular channel. When the spacing between channels exceeds 30 mm, there is almost no change in laminate stiffness, and the variations in tensile and compressive strengths are also relatively minor. However, when the spacing is less than 30 mm, the decrease in the mechanical properties of the laminates becomes significantly more pronounced.

### 4.3. Volume Fraction

The volume fraction of the microvascular channel in the laminate is also an important design parameter in practical applications. Based on the microvascular configuration in this paper, the volume fractions of the microvascular channel corresponding to various channel diameters and spacing are calculated. Figure 15 presents the variation in the mechanical properties of the laminates under specified channel diameters. Since the volume fraction of the microvascular channel with a 0.1 mm diameter is extremely small, making the curve difficult to observe, only the curves with 0.5 mm and 1 mm diameters are presented. Furthermore, as the stiffness of the laminates is minimally affected, the focus is primarily on the variation in strength. It can be observed that the laminate with larger channel diameter has a lower slope on the strength curve. As the volume fraction of the microvascular channel increases, the slope of the strength curves decreases gradually.

### 4.4. Parameter Design Criteria

When designing microvascular self-healing composite structures, it is necessary to consider both the mechanical properties of the laminate and the damage repair capability of the microvascular channel. Generally, to ensure that self-healing is triggered in time before the damage propagation significantly affects the overall performance of the structure, the spacing of the microvascular channel must be less than a certain value. At the same time, there may also be requirements that the mechanical properties of the laminate should not fall below a certain level, or the volume fraction of the microvascular channel should not exceed a certain level when designing structures. So, the variation of the mechanical properties of the microvascular composites can be determined through experiments or finite element methods first, and then the range of microvascular design parameters can be determined based on these limitations. Larger diameter and spacing are better within the allowable design range.

## 5. Conclusions

Tensile and compressive performances of woven fabric CFRP laminates containing three-dimensional microvascular channels were investigated experimentally. Different finite element models with z-direction microvascular channels were established and verified by experimental results. The validated models were used to investigate the damage propagation process and failure mechanism of the laminates under tensile and compressive loads. Then, a parameter study was conducted. The following conclusions can be obtained:(1)The Z-direction microvascular channel has a critical position, which is prone to damage under tensile and compressive loads. It has minimal effect on the stiffness of laminates, but a certain effect on the strength. With a channel diameter of 0.5 mm and a spacing of 12 mm, the tensile and compressive strengths decrease by approximately 10% to 12% compared with blank laminates.(2)The FE models with z-direction microvascular channels which consider different orientations of the resin-rich region formed in the warp and weft fiber yarns agree well with the test results. For woven fabric CFRP composites with z-direction microvascular channels, equivalent unidirectional ply material properties should be assigned separately in an FE model for more accurate calculation, while the effect of variations in the fiber volume fraction around the microvascular channel can be disregarded.(3)Within the common microvascular diameter ranging from 0.1 mm to 1 mm, stiffness variation of the laminates is small, while the laminate strength varies approximately linearly with the channel diameter.(4)The combined investigation using experimental and numerical methods makes it possible to conveniently determine the mechanical properties of laminates with microvascular layers, which provides references for structural design. Moreover, it is also necessary to establish design methods for microvascular parameters in order to apply microvascular composites in engineering structures, which is a key direction for future study.

## Figures and Tables

**Figure 1 polymers-16-00665-f001:**
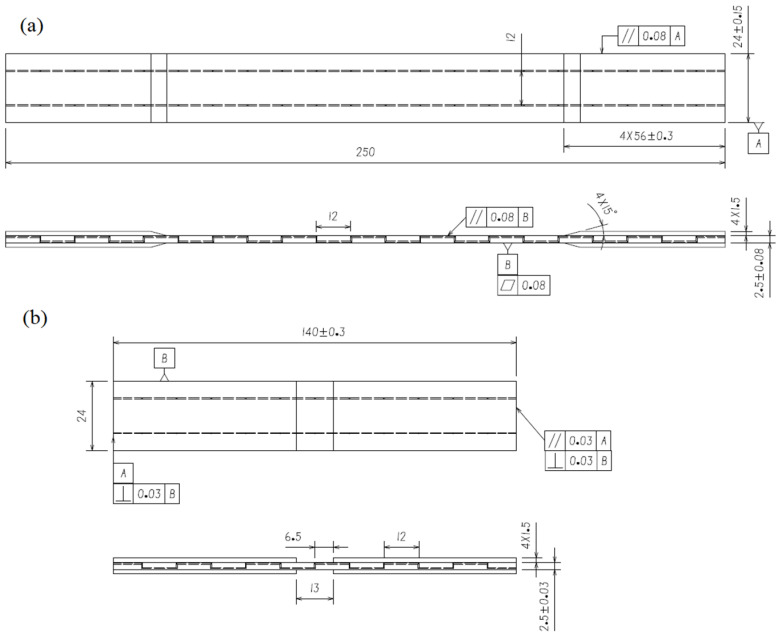
Configuration and geometric parameters (mm) of specimens: tensile specimen (**a**) and compression specimen (**b**).

**Figure 2 polymers-16-00665-f002:**
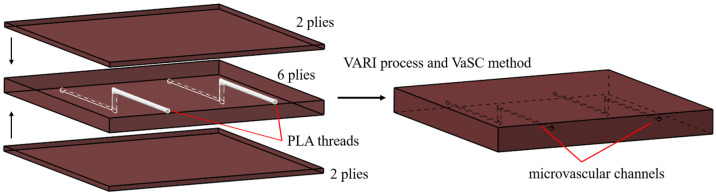
Specimen structure and manufacturing process schematic.

**Figure 3 polymers-16-00665-f003:**
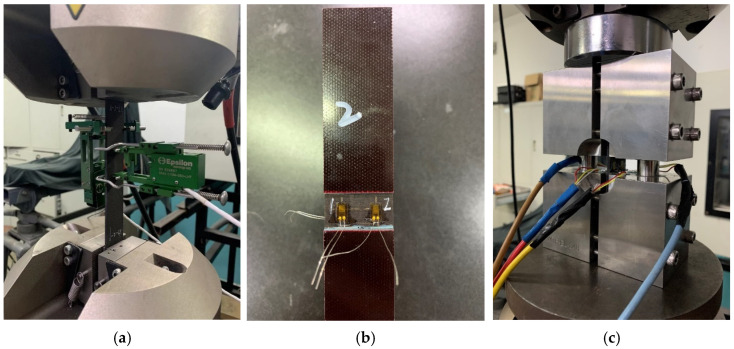
Test set-up and extensometer installation in tensile test (**a**). Strain gauges on compression specimen (**b**). Test set-up in compression test (**c**).

**Figure 4 polymers-16-00665-f004:**
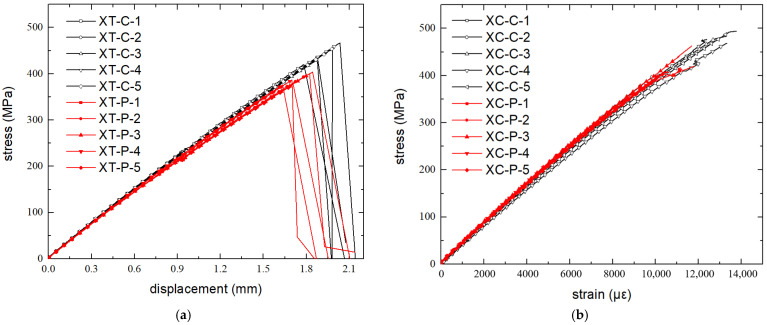
Stress–displacement curves of tensile specimens (**a**) and stress–strain curves of compression specimens (**b**).

**Figure 5 polymers-16-00665-f005:**
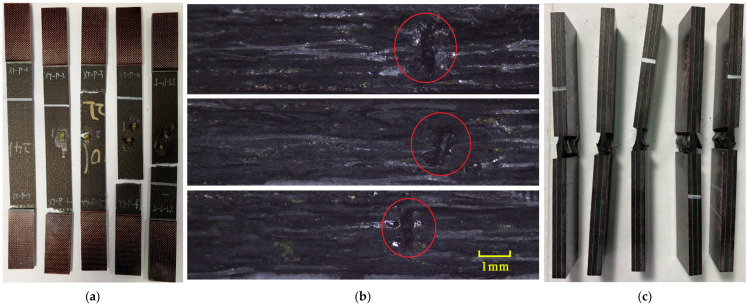
Failure modes of the specimens, lateral tensile failure at middle of gage section or top of grip section (**a**), enlarged view of the tensile fracture section (z-direction channels are circled in red) (**b**), and brooming compression failure at middle of gage section (**c**).

**Figure 6 polymers-16-00665-f006:**
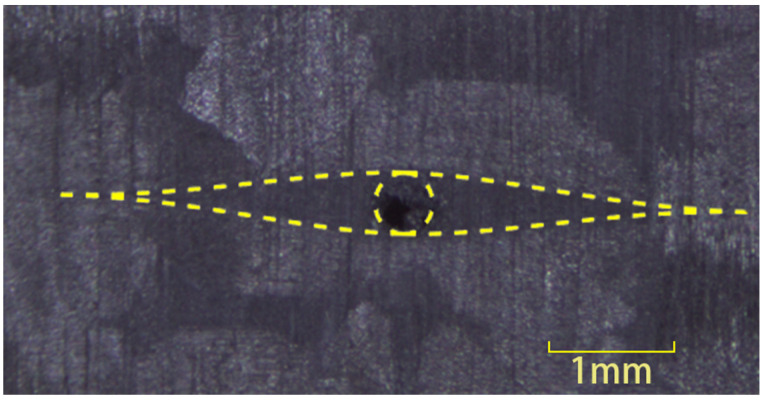
Intralaminar resin-rich region around the microvascular channel. (The boundaries are marked with dotted line).

**Figure 7 polymers-16-00665-f007:**
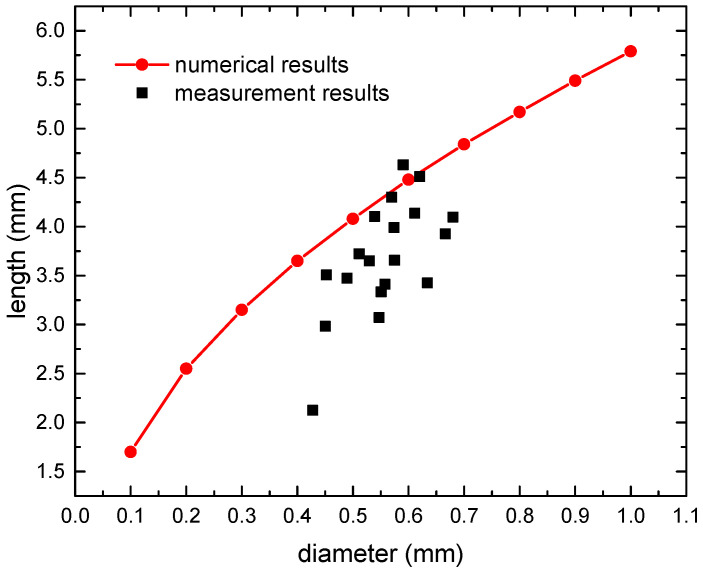
Comparison of numerical and measurement results of resin-rich region length.

**Figure 8 polymers-16-00665-f008:**
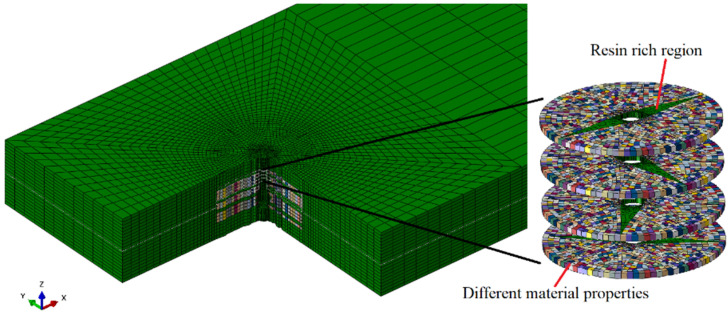
Z-direction microvascular channel in model A (Different colors of elements represent different material properties).

**Figure 9 polymers-16-00665-f009:**
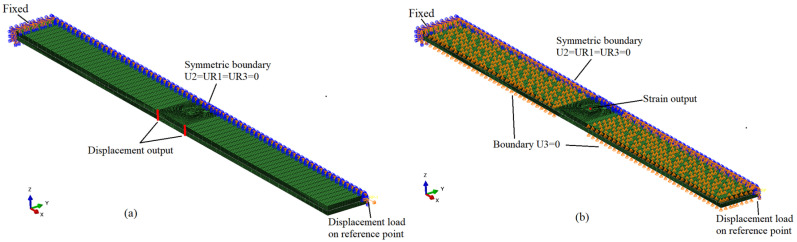
Boundary conditions, loading condition, and nodes for result output: tensile model (**a**) and compression model (**b**).

**Figure 10 polymers-16-00665-f010:**
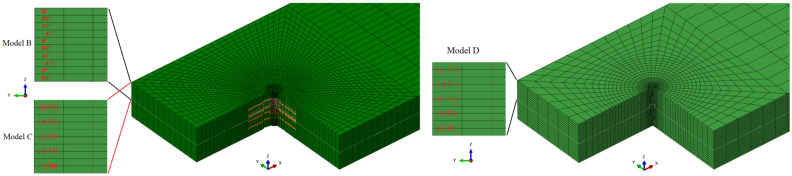
Schematic of the comparison model.

**Figure 11 polymers-16-00665-f011:**
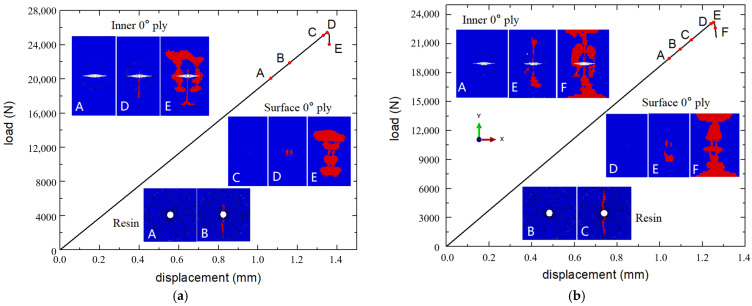
Load–displacement curves and damage propagation of specimens: tensile (**a**) and compression (**b**).

**Figure 12 polymers-16-00665-f012:**
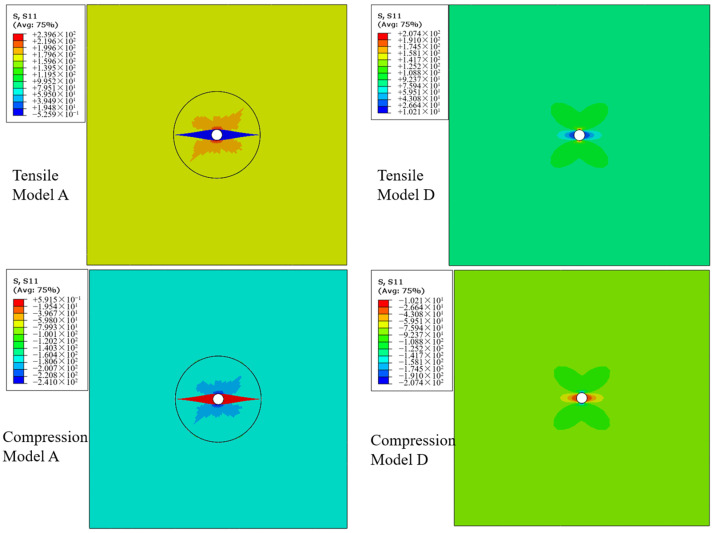
Stress distribution around the microvascular channel.

**Figure 13 polymers-16-00665-f013:**
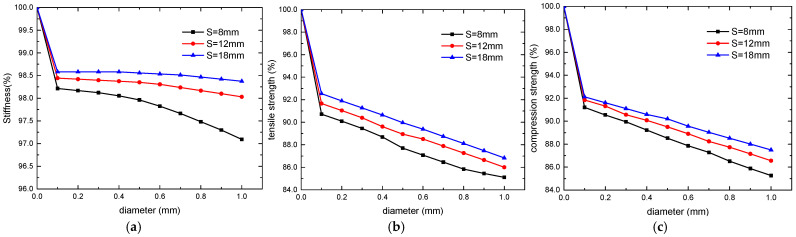
Variation of mechanical properties with diameter of microvascular channels: stiffness (**a**), tensile strength (**b**), and compression strength (**c**).

**Figure 14 polymers-16-00665-f014:**
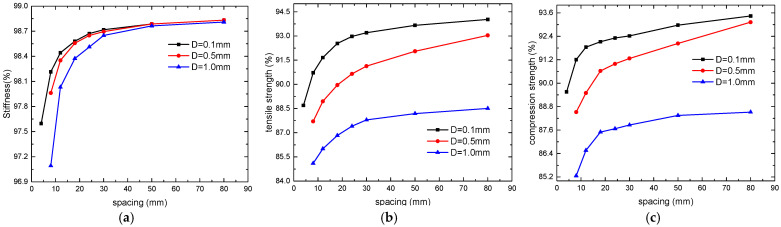
Variation of mechanical properties with spacing of microvascular channel: stiffness (**a**), tensile strength (**b**), and compression strength (**c**).

**Figure 15 polymers-16-00665-f015:**
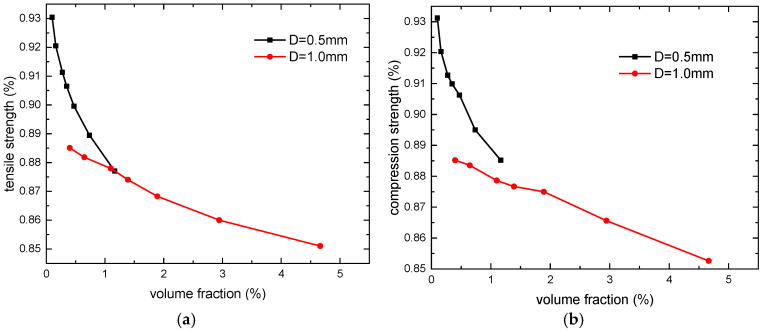
Variation in mechanical properties with volume fraction of microvascular channel: tensile strength (**a**) and compression strength (**b**).

**Table 1 polymers-16-00665-t001:** Mechanical properties of CF3031/5284 ply.

Property	Value	Property	Value
E_1_, E_2_/GPa	55.0	*X*_t_, *Y*_t_/MPa	550
E_3_/GPa	8.4	*X*_c_, *Y*_c_/MPa	593
G_12_/GPa	3.64	*Z*_t_/MPa	80
G_13_, G_23_/GPa	3.0	*Z*_t_/MPa	180
ν_12_	0.051	*S*_12_/MPa	84
ν_13_, ν_23_	0.15	*S*_13_, *S*_23_/MPa	80

**Table 2 polymers-16-00665-t002:** Test results.

Specimen Type	Specimens	Tensile Stiffness/GPa	Tensile Strength/MPa	Specimens	Compression Stiffness/GPa	Compression Strength/MPa
Blank	XT-C-1	43.51	419.75	XC-C-1	41.34	475.67
	XT-C-2	43.36	436.13	XC-C-2	42.22	455.59
	XT-C-3	43.85	430.25	XC-C-3	41.26	477.40
	XT-C-4	45.14	466.70	XC-C-4	39.30	467.92
	XT-C-5	44.05	454.45	XC-C-5	40.40	437.92
	Average	43.98	441.45	Average	40.90	462.9
Microvascular	XT-P-1	41.15	376.07	XC-P-1	41.48	416.11
	XT-P-2	41.46	400.55	XC-P-2	41.20	391.21
	XT-P-3	40.63	404.09	XC-P-3	42.38	438.65
	XT-P-4	41.44	386.48	XC-P-4	42.27	413.13
	XT-P-5	41.72	376.82	XC-P-5	41.45	416.27
	Average	41.30	388.80	Average	41.75	415.10
	Variations of AVG/%	−6.1	−11.9	Variations of AVG/%	2.1	−10.3

**Table 3 polymers-16-00665-t003:** Resin-rich region length in FE model.

Microvascular Diameter/mm	Resin-Rich Region Length/mm	Microvascular Diameter/mm	Resin-Rich Region Length/mm
0.1	1.70	0.6	4.48
0.2	2.55	0.7	4.84
0.3	3.15	0.8	5.17
0.4	3.65	0.9	5.49
0.5	4.04	1.0	5.79

**Table 4 polymers-16-00665-t004:** Mechanical properties of the fiber and resin.

$E_{11}^{f}$/GPa	$E_{22}^{f}$/GPa	$G_{12}^{f}$/GPa	$G_{23}^{f}$/GPa	$\nu_{12}^{f}$	*σ*_f_/MPa	*E*^m^/GPa	*G*^m^/GPa	*ν* ^m^	$S_{t}^{m}$/MPa	$S_{c}^{m}$/MPa
230	13.8	9	4.8	0.2	3530	3.2	1.13	0.42	80	180

**Table 5 polymers-16-00665-t005:** Correction factor value.

*α* _1_	*α* _2_	*α* _3_	*α* _4_	*α* _5_	*β* _t_	*β* _c_
0.8118	0.8257	1.5279	1.1322	1.1538	0.5321	0.5735

**Table 6 polymers-16-00665-t006:** Equivalent unidirectional ply properties calculated by modified Chamis model.

*E*_11_/GPa	*E*_22_/GPa	*ν* _12_	*G*_12_/GPa	*G*_12_/GPa	*X*_t_/MPa	*X*_c_/MPa
103.86	6.14	0.46	3.64	3.00	1038.6	1119.8

**Table 7 polymers-16-00665-t007:** Stiffness degradation rules of composite.

Failure Mode	Stiffness Degradation Rule
Fiber failure	0.07 × all parameters
Matrix failure	0.2 × *E*_22_, *G*_12_, *G*_23_, *μ*_12_, *μ*_23_
Fiber–matrix shear failure	0.2 × *G*_12_, *μ*_12_
Delamination	0.2 × *E*_33_, *G*_13_, *G*_23_, *μ*_13_, *μ*_23_

**Table 8 polymers-16-00665-t008:** Comparison of experiment and simulation results.

Properties	Experiment		Control Model		Model A		Model B		Model C		Model D	
	Control	Vascular	FEM	Error/%	FEM	Error/%	FEM	Error/%	FEM	Error/%	FEM	Error/%
Tensile stiffness/GPa	43.98	41.30	43.67	−0.7	42.95	4.0	42.98	4.1	43.05	4.2	43.29	4.8
Tensile strength/MPa	441.5	388.8	434.7	−1.5	386.6	−0.6	379.1	−2.5	315.1	−19.0	337.7	−13.1
Compression stiffness/GPa	40.90	41.75	43.49	6.3	41.69	−0.1	41.37	−0.9	42.87	2.7	42.86	2.7
Compression strength/MPa	462.9	415.1	469.5	1.4	424.2	2.2	413.1	−0.5	350.5	−15.6	360.4	−13.2

**Table 9 polymers-16-00665-t009:** Experimental and numerical damage configurations.

Load	Model A	Model B	Model C	Model D	Experiment
Tensile	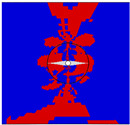	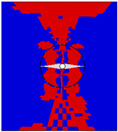	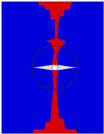	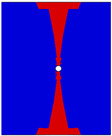	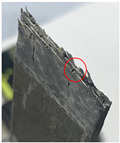
Compression	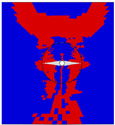	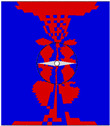	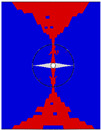	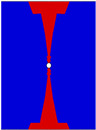	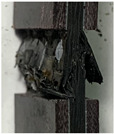

## Data Availability

Data are contained within the article.

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
