# Peer review of "Tensile and Compressive Properties of Woven Fabric Carbon Fiber-Reinforced Polymer Laminates Containing Three-Dimensional Microvascular Channels"

_polymers, 2024, doi:10.3390/polym16050665_

Round 1

Reviewer 1 Report

Comments and Suggestions for Authors

This paper completed experimental and numerical investigations into tensile and compressive properties of woven fabric CFRP laminates. In general, the paper is within the scope of the journal. The research topic is of interest. However, this manuscript cannot be accepted in its current status, needing revisions.

(1)    Introduction. Please clearly state the necessity and novelty of this research.

(2)    Section 2.1. The description of specimen manufacture is suggested to be written in a concise manner.

(3)    Section 2.2. How was the test specimen failure determined during the testing process?

(4)    Please clarify the details and properties of the tested specimens.

(5)    Section 2.3. The stress-strain curves are suggested to be provided and analyzed in detail.

(6)    Section 3.2. How were the parameters for the CFRP identified and used in this model?

(7)    Section 3.1.1. The analysis of the resign rich region is suggested to place after section 3.2.

(8)    How was the mesh size measured and evaluated for the different regions of the specimens?

(9)    Section 3.4. Please compare the tested and simulated damage configurations for these specimens.

(10)What are the limitations and assumptions of the developed models?

Reviewer 2 Report

Comments and Suggestions for Authors

- The Introduction section should be expanded and present a more comprehensive state-of-the-art study;

- Experimental section should include a sub-section describing the materials used in the study (properties and source);

-  The number of specimens per test (mechanical) should be mentioned;

- Figure 4 should describe all three images (only and b are included in the caption text)- also, Figure 4-b should mention how the image was captured (i.e. optical microscopy, high-resolution camera etc) and at what magnification;

-  The manuscript should include information about the novelty/originality of the work as well as the purpose correlated with the target application that the study serves.

- Conclusions should include besides the punctual findings, global observations following the study as well as whether the simulation models can be considered in accordance with experimentally obtained results.

-

Comments on the Quality of English Language

The English language requires some minor corrections
